# Preventing Gradient Explosions in Gated Recurrent Units

**Sekitoshi Kanai, Yasuhiro Fujiwara, Sotetsu Iwamura**
NTT Software Innovation Center
3-9-11, Midori-cho, Musashino-shi, Tokyo
{kanai.sekitoshi, fujiwara.yasuhiro, iwamura.sotetsu}@lab.ntt.co.jp

## Abstract

A gated recurrent unit (GRU) is a successful recurrent neural network architecture for time-series data. The GRU is typically trained using a gradient-based method, which is subject to the exploding gradient problem in which the gradient increases significantly. This problem is caused by an abrupt change in the dynamics of the GRU due to a small variation in the parameters. In this paper, we find a condition under which the dynamics of the GRU changes drastically and propose a learning method to address the exploding gradient problem. Our method constrains the dynamics of the GRU so that it does not drastically change. We evaluated our method in experiments on language modeling and polyphonic music modeling. Our experiments showed that our method can prevent the exploding gradient problem and improve modeling accuracy.

## 1 Introduction

Recurrent neural networks (RNNs) can handle time-series data in many applications such as speech recognition [14, 1], natural language processing [26, 30], and hand writing recognition [13]. Unlike feed-forward neural networks, RNNs have recurrent connections and states to represent the data. Back propagation through time (BPTT) is a standard approach to train RNNs. BPTT propagates the gradient of the cost function with respect to the parameters, such as weight matrices, at each layer and at each time step by unfolding the recurrent connections through time. The parameters are updated using the gradient in a way that minimizes the cost function. The cost function is selected according to the task, such as classification or regression.

Although RNNs are used in many applications, they have problems in that the gradient can be extremely small or large; these problems are called the vanishing gradient and exploding gradient problems [5, 28]. If the gradient is extremely small, RNNs can not learn data with long-term dependencies [5]. On the other hand, if the gradient is extremely large, the gradient moves the RNNs parameters far away and disrupts the learning process. To handle the vanishing gradient problem, previous studies [18, 8] proposed sophisticated models of RNN architectures. One successful model is a long short-term memory (LSTM). However, the LSTM has the complex structures and numerous parameters with which to learn the long-term dependencies. As a way of reducing the number of parameters while avoiding the vanishing gradient problem, a gated recurrent unit (GRU) was proposed in [8]; the GRU has only two gate functions that hold or update the state which summarizes the past information. In addition, Tang et al. [33] show that the GRU is more robust to noise than the LSTM is, and it outperforms the LSTM in several tasks [9, 20, 33, 10].

Gradient clipping is a popular approach to address the exploding gradient problem [26, 28]. This method rescales the gradient so that the norm of the gradient is always less than a threshold. Although gradient clipping is a very simple method and can be used with GRUs, it is heuristic and does not analytically derive the appropriate threshold. The threshold has to be manually tuned to the data

and tasks by trial and error. Therefore, a learning method is required to more effectively address the exploding gradient problem in training of GRUs.

In this paper, we propose a learning method for GRUs that addresses the exploding gradient problem. The method is based on an analysis of the dynamics of GRUs. GRUs suffer from gradient explosions due to their nonlinear dynamics [11, 28, 17, 3] that enable GRUs to represent time-series data. The dynamics can drastically change when the parameters cross certain values, called bifurcation points [36], in the learning process. Therefore, the gradient of the state with respect to the parameters can drastically increase at a bifurcation point. This paper presents an analysis of the dynamics of GRUs and proposes a learning method to prevent the parameters from crossing the bifurcation point. It describes evaluations of this method through language modeling and polyphonic music modeling experiments. The experiments demonstrate that our method can train GRUs without gradient clipping and that it can improve the accuracy of GRUs.

The rest of this paper is organized as follows: Section 2 briefly explains the GRU, dynamical systems and the exploding gradient problem. It also outlines related work. The dynamics of GRUs is analyzed and our training approach is presented in Section 3. The experiments that verified our method are discussed in Section 4. The paper concludes in Section 5. Proofs of lemmas are given in the supplementary material.

## 2 Preliminaries

### 2.1 Gated Recurrent Unit

Time-series data often have long and short-term dependencies. In order to model long and short-term behavior, a GRU is designed to properly keep and forget past information. The GRU controls the past information by having two gates: an update gate and reset gate. The update gate $\boldsymbol{z}_t \in \mathbb{R}^{n \times 1}$ at a time step $t$ is expressed as

$$\boldsymbol{z}_t = \mathrm{sigm}(\boldsymbol{W}_{xz}\boldsymbol{x}_t + \boldsymbol{W}_{hz}\boldsymbol{h}_{t-1}), \tag{1}$$

where $\boldsymbol{x}_t \in \mathbb{R}^{m \times 1}$ is the input vector, and $\boldsymbol{h}_t \in \mathbb{R}^{n \times 1}$ is the state vector. $\boldsymbol{W}_{xz} \in \mathbb{R}^{n \times m}$ and $\boldsymbol{W}_{hz} \in \mathbb{R}^{n \times n}$ are weight matrices. $\mathrm{sigm}(\cdot)$ represents the element-wise logistic sigmoid function. The reset gate $\boldsymbol{r}_t \in \mathbb{R}^{n \times 1}$ is expressed as

$$\boldsymbol{r}_t = \mathrm{sigm}(\boldsymbol{W}_{xr}\boldsymbol{x}_t + \boldsymbol{W}_{hr}\boldsymbol{h}_{t-1}), \tag{2}$$

where $\boldsymbol{W}_{xr} \in \mathbb{R}^{n \times m}$ and $\boldsymbol{W}_{hr} \in \mathbb{R}^{n \times n}$ are weight matrices. The activation of the state $\boldsymbol{h}_t$ is expressed as

$$\boldsymbol{h}_t = \boldsymbol{z}_t \odot \boldsymbol{h}_{t-1} + (\mathbf{1} - \boldsymbol{z}_t) \odot \tilde{\boldsymbol{h}}_t, \tag{3}$$

where $\mathbf{1}$ is the vector of all ones, and $\odot$ means the element-wise product. $\tilde{\boldsymbol{h}}_t$ is a candidate for a new state, expressed as

$$\tilde{\boldsymbol{h}}_t = \tanh(\boldsymbol{W}_{xh}\boldsymbol{x}_t + \boldsymbol{W}_{hh}(\boldsymbol{r}_t \odot \boldsymbol{h}_{t-1})), \tag{4}$$

where $\tanh(\cdot)$ is the element-wise hyperbolic tangent, and $\boldsymbol{W}_{xh} \in \mathbb{R}^{n \times m}$ and $\boldsymbol{W}_{hh} \in \mathbb{R}^{n \times n}$ are weight matrices. The initial value of $\boldsymbol{h}_t$ is $\boldsymbol{h}_0 = \mathbf{0}$ where $\mathbf{0}$ represents the vector of all zeros; the GRU completely forgets the past information when $\boldsymbol{h}_t$ becomes $\mathbf{0}$.

The training of a GRU can be formulated as an optimization problem as follows:

$$\min_{\boldsymbol{\theta}} \tfrac{1}{N} \sum_{j=1}^{N} C(\boldsymbol{x}^{(j)}, \boldsymbol{y}^{(j)}; \boldsymbol{\theta}), \tag{5}$$

where $\boldsymbol{\theta}$, $\boldsymbol{x}^{(j)}$, $\boldsymbol{y}^{(j)}$, $C(\boldsymbol{x}^{(j)}, \boldsymbol{y}^{(j)}; \boldsymbol{\theta})$, and $N$ are all parameters of the model (e.g., elements of $\boldsymbol{W}_{hh}$), the $j$-th training input data, the $j$-th training output data, the loss function for the $j$-th data (e.g., mean squared error or cross entropy), and the number of training data, respectively. This optimization problem is usually solved through stochastic gradient descent (SGD). SGD iteratively updates parameters according to the gradient of a mini-batch, which is randomly sampled data from the training data. The parameter update at step $\tau$ is

$$\boldsymbol{\theta}^{(\tau)} = \boldsymbol{\theta}^{(\tau-1)} - \eta \nabla_{\boldsymbol{\theta}} \tfrac{1}{|D_\tau|} \sum_{(\boldsymbol{x}^{(j)}, \boldsymbol{y}^{(j)}) \in D_\tau} C(\boldsymbol{x}^{(j)}, \boldsymbol{y}^{(j)}; \boldsymbol{\theta}), \tag{6}$$

where $D_\tau$, $|D_\tau|$, and $\eta$ represent the $\tau$-th mini-batch, the size of the mini-batch, and the learning rate of SGD, respectively. In gradient clipping, the norm of $\nabla_{\boldsymbol{\theta}} \tfrac{1}{|D_\tau|} \sum_{(\boldsymbol{x}^{(j)}, \boldsymbol{y}^{(j)}) \in D_\tau} C(\boldsymbol{x}^{(j)}, \boldsymbol{y}^{(j)}; \boldsymbol{\theta})$ is clipped by the specified threshold. The size of the parameters $\boldsymbol{\theta}$ is $3(n^2 + mn) + \alpha$, where $\alpha$ is the number of parameters except for the GRU, because the sizes of the six weight matrices of $\boldsymbol{W}_*$ in eqs. (1)-(4) are $n \times n$ or $n \times m$. Therefore, the computational cost of gradient clipping is $O(n^2 + mn + \alpha)$.

## 2.2 Dynamical System and Gradient Explosion

An RNN is a nonlinear dynamical system that can be represented as follows:

$$\boldsymbol{h}_t = \boldsymbol{f}(\boldsymbol{h}_{t-1}, \boldsymbol{\theta}), \tag{7}$$

where $\boldsymbol{h}_t$ is a state vector at time step $t$, $\boldsymbol{\theta}$ is a parameter vector, and $\boldsymbol{f}$ is a nonlinear vector function. The state evolves over time according to eq. (7). If the state $\boldsymbol{h}_{t_*}$ at some time step $t_*$ satisfies $\boldsymbol{h}_{t_*} = \boldsymbol{f}(\boldsymbol{h}_{t_*}, \boldsymbol{\theta})$, i.e., the new state equals the previous state, the state never changes until an external input is applied to the system. Such a state point is called a fixed point $\boldsymbol{h}_*$. The state converges to or goes away from the fixed point $\boldsymbol{h}_*$ depending on $\boldsymbol{f}$ and $\boldsymbol{\theta}$. This property is important and is called stability [36]. The fixed point $\boldsymbol{h}_*$ is said to be locally stable if there exists a constant $\varepsilon$ such that, for $\boldsymbol{h}_t$ whose initial value $\boldsymbol{h}_0$ satisfies $|\boldsymbol{h}_0 - \boldsymbol{h}_*| < \varepsilon$, $\lim_{t \to \infty} |\boldsymbol{h}_t - \boldsymbol{h}_*| = 0$ holds. In this case, a set of points $\boldsymbol{h}_0$ such that $|\boldsymbol{h}_0 - \boldsymbol{h}_*| < \varepsilon$ is called a basin of attraction of the fixed point. Conversely, if $\boldsymbol{h}_*$ is not stable, the fixed point is said to be unstable. Stability and the behavior of $\boldsymbol{h}_t$ near a fixed point, e.g., converging or diverging, can be qualitatively changed by a smooth variation in $\boldsymbol{\theta}$. This phenomenon is called a local bifurcation, and the value of the parameter of a bifurcation is called a bifurcation point [36].

Doya [11], Pascanu et al. [28] and Baldi and Hornik [3] pointed out that gradient explosions are due to bifurcations. The training of an RNN involves iteratively updating its parameters. This process causes a bifurcation: a small change in parameters can result in a drastic change in the behavior of the state. As a result, the gradient increases at a bifurcation point.

## 2.3 Related Work

Kuan et al. [23] established a learning method to avoid the exploding gradient problem. This method restricts the dynamics of an RNN so that the state remains stable. Yu [37] proposed a learning rate for stable training through Lyapunov functions. However, these methods mainly target Jordan and Elman networks called simple RNNs which, unlike GRUs, are difficult to train long-term dependencies. In addition, they suppose that the mean squared error is used as the loss function. By contrast, our method targets the GRU, a more sophisticated model, and can be used regardless of the loss function. Doya [11] showed that bifurcations cause gradient explosions and that real-time recurrent learning (RTRL) can train an RNN without the gradient explosion. However, RTRL has a high computational cost: $O((n + u)^4)$ for each update step where $u$ is the number of output units [19]. More recently, Arjovsky et al. [2] proposed unitary matrix constraints in order to prevent the gradient vanishing and exploding. Vorontsov et al. [35], however, showed that it can be detrimental to maintain hard constraints on matrix orthogonality.

Previous studies analyzed the dynamics of simple RNNs [12, 4, 31, 16, 27]. Barabanov and Prokhorov [4] analyzed the absolute stability of multi-layer simple RNNs. Haschke and Steil [16] presented a bifurcation analysis of a simple RNN in which inputs are regarded as the bifurcation parameter. Few studies have analyzed the dynamics of the modern RNN models. Talathi and Vartak [32] analyzed the nonlinear dynamics of an RNN with a Relu nonlinearity. Laurent and von Brecht [24] empirically revealed that LSTMs and GRUs can exhibit chaotic behavior and proposed a novel model that has stable dynamics. To the best of our knowledge, our study is the first to analyze the stability of GRUs.

## 3 Proposed Method

As mentioned in Section 2, a bifurcation makes the gradient explode. In this section, through an analysis of the dynamics of GRUs, we devise a training method that avoids a bifurcation and prevents the gradient from exploding.

### 3.1 Formulation of Proposed Training

In Section 3.1, we formulate our training approach. For the sake of clarity, we first explain the formulation for a one-layer GRU; then, we apply the method to a multi-layer GRU.

#### 3.1.1 One-Layer GRU

The training of a GRU is formulated as eq. (5). This training with SGD can be disrupted by a gradient explosion. To prevent the gradient from exploding, we formulate the training of a one-layer GRU as

the following constrained optimization:

$$\min_{\boldsymbol{\theta}} \frac{1}{N} \sum_{j=1}^{N} C(\boldsymbol{x}^{(j)}, \boldsymbol{y}^{(j)}; \boldsymbol{\theta}), \quad \text{s.t.} \quad \sigma_1(\boldsymbol{W}_{hh}) < 2, \tag{8}$$

where $\sigma_i(\cdot)$ is the $i$-th largest singular value of a matrix, and $\sigma_1(\cdot)$ is called the spectral norm. This constrained optimization problem keeps the one-layer GRU locally stable and prevents the gradient from exploding due to a bifurcation of the fixed point on the basis of the following theorem:

**Theorem 1.** *When $\sigma_1(\boldsymbol{W}_{hh}) < 2$, a one-layer GRU is locally stable at a fixed point $\boldsymbol{h}_* = \boldsymbol{0}$.*

We show the proof of this theorem later. This theorem indicates that our training approach of eq. (8) maintains the stability of the fixed point $\boldsymbol{h}_* = \boldsymbol{0}$. Therefore, our approach prevents the gradient explosion caused by the bifurcation of the fixed point $\boldsymbol{h}_*$. In order to prove this theorem, we need to use the following three lemmas:

**Lemma 1.** *A one-layer GRU has a fixed point at $\boldsymbol{h}_* = \boldsymbol{0}$.*

**Lemma 2.** *Let $\boldsymbol{I}$ be an $n \times n$ identity matrix, $\lambda_i(\cdot)$ be the eigenvalue that has the $i$-th largest absolute value, and $\boldsymbol{J} = \frac{1}{4}\boldsymbol{W}_{hh} + \frac{1}{2}\boldsymbol{I}$. When the spectral radius [1] $|\lambda_1(\boldsymbol{J})| < 1$, a one-layer GRU without input can be approximated by the following linearized GRU near $\boldsymbol{h}_t = \boldsymbol{0}$:*

$$\boldsymbol{h}_t = \boldsymbol{J}\boldsymbol{h}_{t-1}, \tag{9}$$

*and the fixed point $\boldsymbol{h}_* = \boldsymbol{0}$ of a one-layer GRU is locally stable.*

Lemma 2 indicates that we can prevent a change in local stability by exploiting the constraint of $|\lambda_1(\frac{1}{4}\boldsymbol{W}_{hh} + \frac{1}{2}\boldsymbol{I})| < 1$. This constraint can be represented as a bilinear matrix inequality (BMI) constraint [7]. However, an optimization problem with a BMI constraint is NP-hard [34]. Therefore, we relax the optimization problem to that of a singular value constraint as in eq. (8) by using the following lemma:

**Lemma 3.** *When $\sigma_1(\boldsymbol{W}_{hh}) < 2$, we have $|\lambda_1(\frac{1}{4}\boldsymbol{W}_{hh} + \frac{1}{2}\boldsymbol{I})| < 1$.*

By exploiting Lemmas 1, 2, and 3, we can prove Theorem 1 as follows:

*Proof.* From Lemma 1, there exists a fixed point $\boldsymbol{h}_* = \boldsymbol{0}$ in a one-layer GRU. This fixed point is locally stable when $|\lambda_1(\frac{1}{4}\boldsymbol{W}_{hh} + \frac{1}{2}\boldsymbol{I})| < 1$ from Lemma 2. From Lemma 3, $|\lambda_1(\frac{1}{4}\boldsymbol{W}_{hh} + \frac{1}{2}\boldsymbol{I})| < 1$ holds when $\sigma_1(\boldsymbol{W}_{hh}) < 2$. Therefore, when $\sigma_1(\boldsymbol{W}_{hh}) < 2$, the one-layer GRU is locally stable at the fixed point $\boldsymbol{h}_* = \boldsymbol{0}$ □

Lemma 1 indicates that a one-layer GRU has a fixed point. Lemma 2 shows the condition under which this fixed point is kept stable. Lemma 3 shows that we can use a singular value constraint instead of an eigenvalue constraint. These lemmas prove Theorem 1, and this theorem ensures that our method prevents the gradient from exploding because of a local bifurcation.

In our method of eq. (8), $\boldsymbol{h}_* = \boldsymbol{0}$ is a fixed point. This fixed point is important since the initial value of the state $\boldsymbol{h}_0$ is $\boldsymbol{0}$, and the GRU forgets all the past information when the state is reset to $\boldsymbol{0}$ as described in Section 2. If $\boldsymbol{h}_* = \boldsymbol{0}$ is stable, the state vector near $\boldsymbol{0}$ asymptotically converges to $\boldsymbol{0}$. This means that the state vector $\boldsymbol{h}_t$ can be reset to $\boldsymbol{0}$ after a sufficient time in the absence of an input; i.e., the GRU can forget the past information entirely. On the other hand, when $|\lambda_1(\boldsymbol{J})|$ becomes greater than one, the fixed point at $\boldsymbol{0}$ becomes unstable. This means that the state vector $\boldsymbol{h}_t$ never resets to $\boldsymbol{0}$; i.e., the GRU can not forget all the past information until we manually reset the state. In this case, the forget gate and reset gate may not work effectively. In addition, Laurent and von Brecht [24] show that an RNN model with state that asymptotically converges to zero achieves a level of performance comparable to that of LSTMs and GRUs. Therefore, our constraint that the GRU is locally stable at $\boldsymbol{h}_* = \boldsymbol{0}$ is effective for learning.

### 3.1.2 Multi-Layer GRU

Here, we extend our method in the multi-layer GRU. An $L$-layer GRU is represented as follows:

$$\boldsymbol{h}_{1,t} = \boldsymbol{f}_1(\boldsymbol{h}_{1,t-1}, \boldsymbol{x}_t), \boldsymbol{h}_{2,t} = \boldsymbol{f}_2(\boldsymbol{h}_{2,t-1}, \boldsymbol{h}_{1,t}), \ldots, \boldsymbol{h}_{L,t} = \boldsymbol{f}_L(\boldsymbol{h}_{L,t-1}, \boldsymbol{h}_{L-1,t}),$$

where $\boldsymbol{h}_{l,t} \in \mathbb{R}^{n_l \times 1}$ is a state vector with the length of $n_l$ at the $l$-th layer, and $\boldsymbol{f}_l$ represents a GRU that corresponds to eqs. (1)-(4) at the $l$-th layer. In the same way as the one-layer GRU, $\boldsymbol{h}_t = [\boldsymbol{h}_{1,t}^{\mathrm{T}}, \ldots, \boldsymbol{h}_{L,t}^{\mathrm{T}}]^{\mathrm{T}} = \boldsymbol{0}$ is a fixed point, and we have the following lemma:

**Lemma 4.** *When $|\lambda_1(\frac{1}{4}\boldsymbol{W}_{l,hh} + \frac{1}{2}\boldsymbol{I})| < 1$ for $l = 1, \ldots, L$, the fixed point $\boldsymbol{h}_* = \boldsymbol{0}$ of a multi-layer GRU is locally stable.*

From Lemma 3, we have $|\lambda_1(\boldsymbol{W}_{l,hh} + \frac{1}{2}\boldsymbol{I})| < 1$ when $\sigma_1(\boldsymbol{W}_{l,hh}) < 2$. Thus, we formulated our training of a multi-layer GRU to prevent gradient explosions as

$$\min_{\boldsymbol{\theta}} \frac{1}{N} \sum_{j=1}^{N} C(\boldsymbol{x}^{(j)}, \boldsymbol{y}^{(j)}; \boldsymbol{\theta}), \ \text{s.t.} \ \sigma_1(\boldsymbol{W}_{l,hh}) < 2, \ \sigma_1(\boldsymbol{W}_{l,xh}) \leq 2 \ \text{for} \ l = 1, \ldots, L. \quad (10)$$

We added the constraint $\sigma_1(\boldsymbol{W}_{l,xh}) \leq 2$ in order to prevent the input from pushing the state out of the basin of attraction of the fixed point $\boldsymbol{h}_* = \boldsymbol{0}$. This constrained optimization problem keeps a multi-layer GRU locally stable.

## 3.2 Algorithm

The optimization method for eq. (8) needs to find the optimal parameters in the feasible set, in which the parameters satisfy the constraint: $\{\boldsymbol{W}_{hh}|\boldsymbol{W}_{hh} \in \mathbb{R}^{n \times n}, \sigma_1(\boldsymbol{W}_{hh}) < 2\}$. Here, we modify SGD in order to solve eq. (8). Our method updates the parameters as follows:

$$\boldsymbol{\theta}_{-\boldsymbol{W}_{hh}}^{(\tau)} = \boldsymbol{\theta}_{-\boldsymbol{W}_{hh}}^{(\tau-1)} - \eta \nabla_{\boldsymbol{\theta}} C_{D_\tau}(\boldsymbol{\theta}), \ \boldsymbol{W}_{hh}^{(\tau)} = \mathcal{P}_\delta(\boldsymbol{W}_{hh}^{(\tau-1)} - \eta \nabla_{\boldsymbol{W}_{hh}} C_{D_\tau}(\boldsymbol{\theta})), \quad (11)$$

where $C_{D_\tau}(\boldsymbol{\theta})$ represents $\frac{1}{|D_\tau|} \sum_{(\boldsymbol{x}^{(j)}, \boldsymbol{y}^{(j)}) \in D_\tau} C(\boldsymbol{x}^{(j)}, \boldsymbol{y}^{(j)}; \boldsymbol{\theta})$, and $\boldsymbol{\theta}_{-\boldsymbol{W}_{hh}}^{(\tau)}$ represents the parameters except for $\boldsymbol{W}_{hh}^{(\tau)}$. In eq. (11), We compute $\mathcal{P}_\delta(\cdot)$ by using the following procedure:

**Step 1.** Decompose $\hat{\boldsymbol{W}}_{hh}^{(\tau)} := \boldsymbol{W}_{hh}^{(\tau-1)} - \eta \nabla_{\boldsymbol{W}_{hh}} C_{D_\tau}(\boldsymbol{\theta})$ by using singular value decomposition (SVD):

$$\hat{\boldsymbol{W}}_{hh}^{(\tau)} = \boldsymbol{U}\boldsymbol{\Sigma}\boldsymbol{V}. \quad (12)$$

**Step 2.** Replace the singular values that are greater than the threshold $2 - \delta$:

$$\bar{\boldsymbol{\Sigma}} = \text{diag}(\min(\sigma_1, 2 - \delta), \ldots \min(\sigma_n, 2 - \delta)). \quad (13)$$

**Step 3.** Reconstruct $\boldsymbol{W}_{hh}^{(\tau)}$ by using $\boldsymbol{U}, \boldsymbol{V}$ and $\bar{\boldsymbol{\Sigma}}$ in Steps 1 and 2:

$$\boldsymbol{W}_{hh}^{(\tau)} \leftarrow \boldsymbol{U}\bar{\boldsymbol{\Sigma}}\boldsymbol{V}. \quad (14)$$

By using this procedure, $\boldsymbol{W}_{hh}$ is guaranteed to have a spectral norm of less than or equal to $2 - \delta$. When $\delta$ is $0 < \delta < 2$, our method constrains $\sigma_1(\boldsymbol{W}_{hh})$ to be less than 2. $\mathcal{P}_\delta(\cdot)$ in our method brings back the parameters into the feasible set when the parameters go out the feasible set after SGD. Our procedure $\mathcal{P}_\delta(\cdot)$ is an optimal projection into the feasible set as shown by the following lemma:

**Lemma 5.** *The weight matrix $\boldsymbol{W}_{hh}^{(\tau)}$ obtained by $\mathcal{P}_\delta(\cdot)$ is a solution of the following optimization:* $\min_{\boldsymbol{W}_{hh}^{(\tau)}} ||\hat{\boldsymbol{W}}_{hh}^{(\tau)} - \boldsymbol{W}_{hh}^{(\tau)}||_F^2, \ \text{s.t.} \ \sigma_1(\boldsymbol{W}_{hh}^{(\tau)}) \leq 2 - \delta,$ *where $|| \cdot ||_F^2$ represents the Frobenius norm.*

Lemma 5 indicates that our method can bring back the weight matrix into the feasible set with minimal variations in the parameters. Therefore, our procedure $\mathcal{P}_\delta(\cdot)$ has minimal impact on the minimization of the loss function. Note that our method does not depend on the learning rate schedule, and an adaptive learning rate method (such as Adam [21]) can be used with it.

## 3.3 Computational Cost

Let $n$ be the length of a state vector $\boldsymbol{h}_t$; a naive implementation of SVD needs $O(n^3)$ time. Here, we propose an efficient method to reduce the computational cost. First, let us reconsider the computation of $\mathcal{P}_\delta(\cdot)$. Equations (12)-(14) can be represented as follows:

$$\boldsymbol{W}_{hh}^{(\tau)} = \hat{\boldsymbol{W}}_{hh}^{(\tau)} - \sum_{i=1}^{s} \left[ \sigma_i(\hat{\boldsymbol{W}}_{hh}^{(\tau)}) - (2 - \delta) \right] \boldsymbol{u}_i \boldsymbol{v}_i^T, \quad (15)$$

where $s$ is the number of the singular values greater than $2 - \delta$, and $\boldsymbol{u}_i$ and $\boldsymbol{v}_i$ are the $i$-th left and right singular vectors, respectively. Eq. (15) shows that our method only needs the singular values and vectors such that $\sigma_i(\hat{\boldsymbol{W}}_{hh}^{(\tau)}) > 2 - \delta$. In order to reduce the computational cost of our method, we use the truncated SVD [15] to efficiently compute the top $s$ singular values in $O(n^2 \log(s))$ time, where $s$ is the specified number of singular values. Since the truncated SVD requires $s$ to be set beforehand, we need to efficiently estimate the number of singular values such that must meet the condition of $\sigma_i(\hat{\boldsymbol{W}}_{hh}^{(\tau)}) > 2 - \delta$. Therefore, we compute upper bounds of the singular values that meet the condition on the basis of the following lemma:

**Lemma 6.** *The singular values of $\hat{\boldsymbol{W}}_{hh}^{(\tau)}$ are bounded with the following inequality:* $\sigma_i(\hat{\boldsymbol{W}}_{hh}^{(\tau)}) \leq \sigma_i(\boldsymbol{W}_{hh}^{(\tau-1)}) + |\eta| ||\nabla_{\boldsymbol{W}_{hh}} C_{D_\tau}(\boldsymbol{\theta})||_F$.

Using this upper bound, we can estimate $s$ as the number of the singular values with upper bounds of greater than $2-\delta$. This upper bound can be computed in $O(n^2)$ time since the size of $\nabla_{\boldsymbol{W}_{hh}} C_{D_\tau}(\boldsymbol{\theta})$ is $n \times n$ and $\sigma_i(\boldsymbol{W}_{hh}^{(\tau-1)})$ has already been obtained at step $\tau$. If we did not compute the previous singular values from $\tau - K$ step to $\tau - 1$ step, we compute the upper bound of $\sigma_i(\hat{\boldsymbol{W}}_{hh}^{(\tau)})$ as $\sigma_i(\boldsymbol{W}_{hh}^{(\tau-K-1)}) + \sum_{k=0}^{K} |\eta| ||\nabla_{\boldsymbol{W}_{hh}} C_{D_{\tau-k}}(\boldsymbol{\theta})||_F$ from Lemma 6. Since our training originally constrains $\sigma_1(\boldsymbol{W}_{hh}^{(\tau)}) < 2$ as described in eq. (8), we can redefine $s$ as the number of singular values such that $\sigma_i(\hat{\boldsymbol{W}}_{hh}^{(\tau)}) > 2$, instead of $\sigma_i(\hat{\boldsymbol{W}}_{hh}^{(\tau)}) > 2 - \delta$. This modification can further reduce the computational cost without disrupting the training. In summary, our method can efficiently estimate the number of singular values needed in $O(n^2)$ time, and we compute the truncated SVD in $O(n^2 \log(s))$ time only if we need to compute singular values by using Lemma 6.

## 4 Experiments

### 4.1 Experimental Conditions

To evaluate the effectiveness of our method, we conducted experiments on language modeling and polyphonic music modeling. We trained the GRU and examined the successful training rate, as well as the average and standard deviation of the loss. We defined successful training as training in which the validation loss at each epoch is never greater than the initial value. The experimental conditions of each modeling are explained below.

#### 4.1.1 Language Modeling

Penn Treebank (PTB) [25] is a widely used dataset to evaluate the performance of RNNs. PTB is split into training, validation, and test sets, and the sets are composed of 930 k, 74 k, 80 k tokens. This experiment used a 10 k word vocabulary, and all words outside the vocabulary were mapped to a special token. The experimental conditions were based on the previous paper [38]. Our model architecture was as follows: The first layer was a $650 \times 10,000$ linear layer without bias to convert the one-hot vector input into a dense vector, and we multiplied the output of the first layer by 0.01 because our method assumes small inputs. The second layer was a GRU layer with 650 units, and we used the softmax function as the output layer. We applied 50 % dropout to the output of each layer except for the recurrent connection [38]. We unfolded the GRU for 35 time steps in BPTT and set the mini-batch size to 20. We trained the GRU with SGD for 75 epochs since the performance of the models trained by Adam and RMSprop were worse than that trained by SGD in the preliminary experiments, and Zaremba et al. [38] used SGD. The results and conditions of preliminary experiments are in the supplementary material. We set the learning rate to one in the first 10 epochs, and then, divided the learning rate by 1.1 after each epoch. In our method, $\delta$ was set to [0.2, 0.5, 0.8, 1.1, 1.4]. In gradient clipping, a heuristic for setting the threshold is to look at the average norm of the gradient [28]. We evaluated gradient clipping based on the gradient norm by following the study [28]. In the supplementary material, we evaluated gradient elementwise clipping which is used practically. Since the average norm of the gradient was about 10, we set the threshold to [5, 10, 15, 20]. We initialized the weight matrices except for $\boldsymbol{W}_{hh}$ with a normal distribution $\mathcal{N}(0, 1/650)$, and $\boldsymbol{W}_{hh}$ as an orthogonal matrix composed of the left singular vectors of a random matrix [29, 8]. After each epoch, we evaluated the validation loss. The model that achieved the least validation loss was evaluated using the test set.

#### 4.1.2 Polyphonic Music Modeling

In this modeling, we predicted MIDI note numbers at the next time step given the observed notes of the previous time steps. We used the Nottingham dataset: a MIDI file containing 1200 folk tunes [6]. We represented the notes at each time step as a 93-dimensional binary vector. This dataset is split into training, validation and test sets [6]. The experimental conditions were based on the previous study [20]. Our model architecture was as follows: The first layer was a $200 \times 93$ linear layer without bias, and the output of the first layer was multiplied by 0.01. The second and third layers were GRU

Table 1: Language modeling results: success rate and perplexity.

| Our method | | | | | | Gradient clipping | | | | |
|---|---|---|---|---|---|---|---|---|---|---|
| Delta | 0.2 | 0.5 | 0.8 | 1.1 | 1.4 | Threshold | 5 | 10 | 15 | 20 |
| Success Rate | 100 % | 100 % | 100 % | 100 % | 100 % | Success Rate | 100 % | 40 % | 0 % | 0 % |
| Validation Loss | 102.0±0.3 | 102.8±0.3 | 103.7±0.2 | 105.2±0.2 | 107.0±0.4 | Validation Loss | 109.3±0.4 | 103.1±0.4 | N/A | N/A |
| Test Loss | 97.6±0.4 | 98.4±0.3 | 99.0±0.4 | 100.3±0.2 | 102.1±0.2 | Test Loss | 106.9±0.4 | 100.4±0.5 | N/A | N/A |

Table 2: Music modeling results: success rate and negative log-likelihood.

| Our method | | | | | | Gradient clipping | | | | |
|---|---|---|---|---|---|---|---|---|---|---|
| Delta | 0.2 | 0.5 | 0.8 | 1.1 | 1.4 | Threshold | 15 | 30 | 45 | 60 |
| Success Rate | 100 % | 100 % | 100 % | 100 % | 100 % | Success Rate | 100 % | 100 % | 100 % | 100 % |
| Validation Loss | 3.46±0.05 | 3.47±0.07 | 3.59±0.1 | 4.58±0.2 | 4.64±0.2 | Validation Loss | 3.57±0.01 | 3.61±0.2 | 3.88±0.2 | 5.26±3 |
| Test Loss | 3.53±0.04 | 3.53±0.04 | 3.64±0.2 | 4.56±0.2 | 4.62±0.2 | Test Loss | 3.64±0.04 | 3.64±0.2 | 3.89±0.2 | 5.36±3 |

layers with 200 units per layer, and we used the logistic function as the output layer. 50 % dropout was applied to non-recurrent connections. We unfolded GRUs for 35 time steps and set the size of the mini-batch to 20. We used SGD with a learning rate of 0.1 and divided the learning rate by 1.25 if we observed no improvement over 10 consecutive epochs. We repeated the same procedure until the learning rate became smaller than $10^{-4}$. In our method, $\delta$ was set to [0.2, 0.5, 0.8, 1.1, 1.4]. In gradient clipping, the threshold was set to [15, 30, 45, 60], since the average norm of the gradient was about 30. We initialized the weight matrices except for $\boldsymbol{W}_{hh}$ with a normal distribution $\mathcal{N}(0, 10^{-4}/200)$ , and $\boldsymbol{W}_{hh}$ as an orthogonal matrix. After each epoch, we evaluated the validation loss, and the model that achieved the least validation loss was evaluated using the test set.

## 4.2 Success Rate and Accuracy

Tables 1 and 2 list the success rates of language modeling and music modeling, respectively. These tables also list the averages and standard deviations of the loss in each modeling to show that our method outperforms gradient clipping. In these tables, "Threshold" means the threshold of gradient clipping, and "Delta" means $\delta$ in our method.

As shown in Table 1, in language modeling, gradient clipping failed to train even though its parameter was set to 10, which is the average norm of the gradient as recommended by Pascanu et al. [28]. Although gradient clipping successfully trained the GRU when its threshold was five, it failed to effectively learn the model with this setting; a threshold of 10 achieved lower perplexity than a threshold of five. As shown in Table 2, in music modeling, gradient clipping successfully trained the GRU. However, the standard deviation of the loss was high when the threshold was set to 60 (double the average norm). On the other hand, our method successfully trained the GRU in both modelings. Tables 1 and 2 show that our approach achieved lower perplexity and negative log-likelihood compared with gradient clipping, while it constrained the GRU to be locally stable. This is because our approach of constraining stability improves the performance of the GRU. The previous study [22] showed that stabilizing the activation of the RNN can improve performance on several tasks. In addition, Bengio et al. [5] showed that an RNN is robust to noise when the state remains in the basin of attraction. Using our method, the state of the GRU tends to remain in the basin of the attraction of $\boldsymbol{h}_* = \boldsymbol{0}$. Therefore, our method can improve robustness against noise, which is an advantage of the GRU [33].

As shown in Table 2, when $\delta$ was set to 1.1 or 1.4, the performance of the GRU deteriorated. This is because the convergence speed of the state depends on $\delta$. As mentioned in Section 3.2, the spectral norm of $\boldsymbol{W}_{hh}$ is less than or equal to $2 - \delta$. This spectral norm gives the upper bound of $|\lambda_1(\boldsymbol{J})|$. $|\lambda_1(\boldsymbol{J})|$ gives the rate of convergence of a linearized GRU (eq. (9)), which approximates GRU near $\boldsymbol{h}_t = \boldsymbol{0}$ when $|\lambda_1(\boldsymbol{J})| < 1$. Therefore, the state of the GRU near $\boldsymbol{h}_t = \boldsymbol{0}$ tends to converge quickly if $\delta$ is set to close to two. In this case, the GRU becomes robust to noise since the state affected by the past noise converges to zero quickly, while the GRU loses effectiveness for long-term dependencies. We can tune $\delta$ from the characteristics of the data: if data have the long-term dependencies, we should set $\delta$ small, whereas we should set $\delta$ large for noisy data.

The threshold in gradient clipping is unbounded, and hence, it is difficult to tune. Although the threshold can be heuristically set on the basis of the average norm, this may not be effective in language modeling using the GRU, as shown in Table 1. In contrast, the hyper-parameter is bounded in our method, i.e., $0 < \delta < 2$, and it is easy to understand its effect as mentioned above.

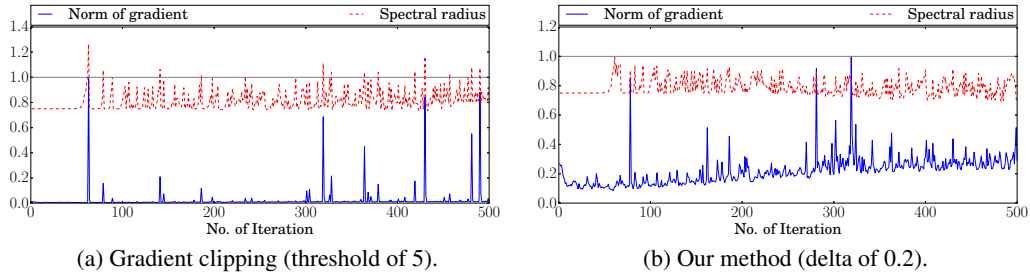

(a) Gradient clipping (threshold of 5).  (b) Our method (delta of 0.2).
Figure 1: Gradient explosion in language modeling.

Table 3: Computation time in the language modeling (delta is 0.2, threshold is 5).

| | Computation time (s) | |
| Naive SVD | Truncated SVD | Gradient clipping |
| --- | --- | --- |
| $5.02 \times 10^4$ | $4.55 \times 10^4$ | $4.96 \times 10^4$ |

### 4.3 Relation between Gradient and Spectral Radius

Our method of constraining the GRU to be locally stable is based on the hypothesis that a change in stability causes an exploding gradient problem. To confirm this hypothesis, we examined (i) the norm of the gradient before clipping and (ii) the spectral radius of $J$ (in Lemma 2), which determines local stability, versus the number of iterations until the 500th iteration in Fig. 1. Fig. 1(a) and 1(b) show the results of gradient clipping with a threshold of 5 and our method with $\delta$ of 0.2. Each norm of the gradient was normalized so that its maximum value was one. The norm of the gradient significantly increased when the spectral radius crossed one, such as at the 63rd, 79th, and 141st iteration (Fig. 1(a)). In addition, the spectral radius decreased to less than one after the gradient explosion; i.e., when the gradient explosion occurred, the gradient became in the direction of decreasing spectral radius. In contrast, our method kept the spectral radius less than one by constraining the spectral norm of $W_{hh}$ (Fig. 1(b)). Therefore, our method can prevent the gradient from exploding and effectively train the GRU.

### 4.4 Computation Time

We evaluated computation time of the language modeling experiment. The detailed experimental setup is described in the supplementary material. Table 3 lists the computation time of the whole learning process using gradient clipping and our method with the naive SVD and with truncated SVD. This table shows the computation time of our method is comparable to gradient clipping. As mentioned in Section 2.1, the computational cost of gradient clipping is proportional to the number of parameters including weight matrices of input and output layers. In language modeling, the sizes of input and output layers tend to be large due to the large vocabulary size. On the other hand, the computational cost of our method only depends on the length of the state vector, and our method can be efficiently computed if the number of singular values greater than 2 is small as described in Section 3.3. As a result, our method could reduce the computation time comparing to gradient clipping.

## 5 Conclusion

We analyzed the dynamics of GRUs and devised a learning method that prevents the exploding gradient problem. Our analysis of stability provides new insight into the behavior of GRUs. Our method constrains GRUs so that the states near $\mathbf{0}$ asymptotically converge to $\mathbf{0}$. Through language and music modeling experiments, we confirmed that our method can successfully train GRUs and found that our method can improve their performance.

## Footnotes

[1]The spectral radius is the maximum value of the absolute value of the eigenvalues.

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
