[Supplementary Material]

# Supplementary Material for Preventing Gradient Explosions in Gated Recurrent Units

## A Proofs of lemmas

First, we show GRU as follows:

$$z_t = \text{sigm}(W_{xz}x_t + W_{hz}h_{t-1}), \tag{1}$$

$$r_t = \text{sigm}(W_{xr}x_t + W_{hr}h_{t-1}), \tag{2}$$

$$h_t = z_t \odot h_{t-1} + (1 - z_t) \odot \tilde{h}_t, \tag{3}$$

$$\tilde{h}_t = \tanh(W_{xh}x_t + W_{hh}(r_t \odot h_{t-1})), \tag{4}$$

where $x_t \in \mathbb{R}^{m \times 1}$ is the input vector, and $h_t \in \mathbb{R}^{n \times 1}$ is the state vector. $W_{xz} \in \mathbb{R}^{n \times m}$, $W_{hz} \in \mathbb{R}^{n \times n}$, $W_{xr} \in \mathbb{R}^{n \times m}$, $W_{hr} \in \mathbb{R}^{n \times n}$ $W_{xh} \in \mathbb{R}^{n \times m}$, and $W_{hh} \in \mathbb{R}^{n \times n}$ are weight matrices. $\text{sigm}(\cdot)$ and $\tanh(\cdot)$ represent the element-wise logistic sigmoid function and hyperbolic tangent, respectively. $1$ is the vector of all ones, and $\odot$ means the element-wise product.

### A.1 Proof of Lemma 1

**Lemma 1.** *A one-layer GRU has a fixed point at $h_* = 0$.*

*Proof.* Substituting an input vector $x_t = 0$ and a previous state vector $h_{t-1} = 0$ in eqs. (1) and (2), we get an update gate $z_t = \frac{1}{2}$ and reset gate $r_t = \frac{1}{2}$. Then, substituting $x_t = 0$, $h_{t-1} = 0$, $r_t = \frac{1}{2}$ in eq. (4) gives a candidate state $\tilde{h}_t = 0$. Finally, by substituting $h_{t-1} = 0$, $z_t = \frac{1}{2}$, $\tilde{h}_t = 0$ for eq. (3), we get a new state $h_t = 0$. Therefore, $h_{t-1} = h_t = 0$ holds, and thus, GRU has a fixed point $h_* = 0$. $\square$

### A.2 Proof of Lemma 2

**Lemma 2.** *Let $I$ be an $n \times n$ identity matrix, $\lambda_i(\cdot)$ be the eigenvalue that has the $i$-th largest absolute value, and $J = \frac{1}{4}W_{hh} + \frac{1}{2}I$. When the spectral radius $|\lambda_1(J)| < 1$, a one-layer GRU without input can be approximated by the following linearized GRU near $h_t = 0$:*

$$h_t = Jh_{t-1}, \tag{5}$$

*and the fixed point $h_* = 0$ of a one-layer GRU is locally stable.*

*Proof.* Local stability can be revealed through the analysis of a linearized system at a fixed point [8]. A linearized GRU at $h_* = 0$ without an input can be obtained from a first-order Taylor expansion [5], as follows:

$$h_t = Jh_{t-1}. \tag{6}$$

In eq. (6), $J$ is the Jacobian matrix of $h_t$ with respect to the state $h_{t-1}$ at $h_{t-1} = 0$ and $x_t = 0$; $J$ is obtained as follows:

$$J = \left. \frac{\partial h_t}{\partial h_{t-1}} \right|_{h_{t-1}=0, x_t=0} = \frac{1}{4}W_{hh} + \frac{1}{2}I. \tag{7}$$

From eq. (6), we have

$$h_t = J^t h_0. \tag{8}$$

Since the $J^t$ depends on the $t$-th powers of the eigenvalues of $J$, eq. (8) indicates that the eigenvalues of $J$ determine the behavior of the linearized GRU. From the Hartman-Grobman theorem, the behavior of a dynamical system near a hyperbolic fixed point is homeomorphic to the behavior of the linearized system [8]. When $|\lambda_1(J)| < 1$, the fixed point $h_* = 0$ is a hyperbolic fixed point. Therefore, a one-layer GRU without input can be approximated as eq. (6). Then, local stability is determined by the spectral radius $|\lambda_1(J)|$. When $|\lambda_1(J)| < 1$, we have $\lim_{t \to \infty} |h_t - h_*| = \lim_{t \to \infty} |h_t| = \lim_{t \to \infty} |J^t h_0| = 0$ for $h_t$ whose initial value $h_0$ is sufficiently near $h_* = 0$. As a result, the fixed point $h_* = 0$ is locally stable. $\square$

## A.3 Proof of Lemma 3

**Lemma 3.** *When $\sigma_1(\boldsymbol{W}_{hh}) < 2$, we have $|\lambda_1(\frac{1}{4}\boldsymbol{W}_{hh} + \frac{1}{2}\boldsymbol{I})| < 1$.*

*Proof.* Let $\boldsymbol{M}$ be an $n \times n$ square matrix and $c$ be a scalar value; the eigenvalue of $\boldsymbol{M} + c\boldsymbol{I}$ becomes $\lambda_i(\boldsymbol{M}) + c$. Therefore, $|\lambda_1(\frac{1}{4}\boldsymbol{W}_{hh} + \frac{1}{2}\boldsymbol{I})| = |\frac{1}{4}\lambda_1(\boldsymbol{W}_{hh}) + \frac{1}{2}|$ holds. From the triangle inequality, $|\frac{1}{4}\lambda_1(\boldsymbol{W}_{hh}) + \frac{1}{2}| \leq \frac{1}{4}|\lambda_1(\boldsymbol{W}_{hh})| + \frac{1}{2}$. We have Weyl's inequality for its singular values and eigenvalues: $\sum_{i=1}^{k}|\lambda_i(\boldsymbol{M})| \leq \sum_{i=1}^{k}\sigma_i(\boldsymbol{M})$ for $k = 1, 2, \ldots, n$. Therefore, $|\lambda_1(\boldsymbol{W}_{hh})| \leq \sigma_1(\boldsymbol{W}_{hh})$ and $\frac{1}{4}|\lambda_1(\boldsymbol{W}_{hh})| + \frac{1}{2}$ is less than or equal to $\frac{1}{4}\sigma_1(\boldsymbol{W}_{hh}) + \frac{1}{2}$. As a result, when $\sigma_1(\boldsymbol{W}_{hh}) < 2$, we have $1 > \frac{1}{4}\sigma_1(\boldsymbol{W}_{hh}) + \frac{1}{2} \geq \frac{1}{4}|\lambda_1(\boldsymbol{W}_{hh})| + \frac{1}{2} \geq |\lambda_1(\frac{1}{4}\boldsymbol{W}_{hh} + \frac{1}{2}\boldsymbol{I})|$. Therefore, we have $|\lambda_1(\frac{1}{4}\boldsymbol{W}_{hh} + \frac{1}{2}\boldsymbol{I})| < 1$ when $\sigma_1(\boldsymbol{W}_{hh}) < 2$. $\qquad\square$

## A.4 Proof of Lemma 4

**Lemma 4.** *When $|\lambda_1(\frac{1}{4}\boldsymbol{W}_{l,hh} + \frac{1}{2}\boldsymbol{I})| < 1$ for $l = 1, \ldots, L$, the fixed point $\boldsymbol{h}_* = \boldsymbol{0}$ of a multi-layer GRU is locally stable.*

*Proof.* In the same way as the one-layer GRU, $\boldsymbol{h}_t = [\boldsymbol{h}_{1,t}^{\mathrm{T}}, \ldots, \boldsymbol{h}_{L,t}^{\mathrm{T}}]^{\mathrm{T}} = \boldsymbol{0}$ is a fixed point, and the Jacobian matrix $\frac{\partial \boldsymbol{h}_t}{\partial \boldsymbol{h}_{t-1}}$ at $\boldsymbol{h}_{t-1} = \boldsymbol{0}$ and $\boldsymbol{x}_t = \boldsymbol{0}$ becomes

$$
\boldsymbol{J} = \begin{bmatrix} \frac{\partial \boldsymbol{h}_{1,t}}{\partial \boldsymbol{h}_{1,t-1}} & \boldsymbol{O} & \cdots & & \boldsymbol{O} \\ \frac{\partial \boldsymbol{h}_{2,t}}{\partial \boldsymbol{h}_{1,t-1}} & \frac{\partial \boldsymbol{h}_{2,t}}{\partial \boldsymbol{h}_{2,t-1}} & \boldsymbol{O} & \cdots & \vdots \\ \vdots & \ddots & \ddots & \ddots & \\ \frac{\partial \boldsymbol{h}_{L-1,t}}{\partial \boldsymbol{h}_{1,t-1}} & \frac{\partial \boldsymbol{h}_{L-1,t}}{\partial \boldsymbol{h}_{2,t-1}} & \cdots & \frac{\partial \boldsymbol{h}_{L-1,t}}{\partial \boldsymbol{h}_{L-1,t-1}} & \boldsymbol{O} \\ \frac{\partial \boldsymbol{h}_{L,t}}{\partial \boldsymbol{h}_{1,t-1}} & \frac{\partial \boldsymbol{h}_{L,t}}{\partial \boldsymbol{h}_{2,t-1}} & \cdots & \frac{\partial \boldsymbol{h}_{L,t}}{\partial \boldsymbol{h}_{L-1,t-1}} & \frac{\partial \boldsymbol{h}_{L,t}}{\partial \boldsymbol{h}_{L,t-1}} \end{bmatrix}. \tag{9}
$$

This matrix is a block lower triangular matrix. The diagonal and off-diagonal blocks respectively become

$$\frac{\partial \boldsymbol{h}_{l,t}}{\partial \boldsymbol{h}_{l,t-1}} = \boldsymbol{A}_l, \tag{10}$$

$$\frac{\partial \boldsymbol{h}_{l,t}}{\partial \boldsymbol{h}_{l-d,t-1}} = \sum_{p=0}^{d}\left[\prod_{k=0}^{p-1}(\boldsymbol{B}_{l-k})\boldsymbol{A}_{l-p}\prod_{q=0}^{-p+d-1}(\boldsymbol{B}_{l-p-q})\right], \tag{11}$$

where $\boldsymbol{A}_l = \frac{1}{4}\boldsymbol{W}_{l,hh} + \frac{1}{2}\boldsymbol{I}$, $\boldsymbol{B}_l = \frac{1}{2}\boldsymbol{W}_{l,xh}$, $\boldsymbol{W}_{l,hh}$ and $\boldsymbol{W}_{l,xh}$ are weight matrices of the GRU at the $l$-th layer. Because the eigenvalues of a block lower triangular matrix correspond to the eigenvalues of diagonal blocks, the eigenvalues of $\boldsymbol{J}$ can be obtained from those of $\frac{\partial \boldsymbol{h}_{l,t}}{\partial \boldsymbol{h}_{l,t-1}}$ for $l = 1, \ldots, L$. Therefore, the spectral radius becomes

$$|\lambda_1(\boldsymbol{J})| = \max_l |\lambda_1(\tfrac{1}{4}\boldsymbol{W}_{l,hh} + \tfrac{1}{2}\boldsymbol{I})|. \tag{12}$$

As a result, when $|\lambda_1(\frac{1}{4}\boldsymbol{W}_{l,hh} + \frac{1}{2}\boldsymbol{I})| < 1$ for $l = 1, \ldots, L$, $|\lambda_1(\boldsymbol{J})| = \max_l |\lambda_1(\frac{1}{4}\boldsymbol{W}_{l,hh} + \frac{1}{2}\boldsymbol{I})| < 1$. $\qquad\square$

## A.5 Proof of Lemma 5

**Lemma 5.** *The weight matrix $\boldsymbol{W}_{hh}^{(\tau)}$ obtained by $\mathcal{P}_\delta(\cdot)$ in our method is a solution of the following optimization:*

$$\min_{\boldsymbol{W}_{hh}^{(\tau)}}||\hat{\boldsymbol{W}}_{hh}^{(\tau)} - \boldsymbol{W}_{hh}^{(\tau)}||_F^2, \ \ \text{s.t. } \sigma_1(\boldsymbol{W}_{hh}^{(\tau)}) \leq 2 - \delta, \tag{13}$$

*where $||\cdot||_F^2$ represents the Frobenius norm.*

*Proof.* The following inequality holds for any square matrices $\boldsymbol{M} \in \mathbb{R}^{q \times q}$ and $\boldsymbol{K} \in \mathbb{R}^{q \times q}$ [2]:

$$\sum_{i=1}^{q}\{\sigma_i(\boldsymbol{M}) - \sigma_i(\boldsymbol{K})\}^2 \leq ||\boldsymbol{M} - \boldsymbol{K}||_F^2, \tag{14}$$

Equation (14) gives $||\hat{\boldsymbol{W}}_{hh}^{(\tau)} - \boldsymbol{W}_{hh}^{(\tau)}||_F^2 \geq \sum_{i=1}^{n}\{\sigma_i(\hat{\boldsymbol{W}}_{hh}^{(\tau)}) - \sigma_i(\boldsymbol{W}_{hh}^{(\tau)})\}^2$. In addition, the lower bound of the right hand side is $\sum_{i=1}^{n}\{\sigma_i(\hat{\boldsymbol{W}}_{hh}^{(\tau)}) - \sigma_i(\boldsymbol{W}_{hh}^{(\tau)})\}^2 \geq \sum_{i=1}^{s}\{(\sigma_i(\hat{\boldsymbol{W}}_{hh}^{(\tau)}) - (2-\delta)\}^2$, where

$s$ is the number of singular values that are greater than $2 - \delta$. On the other hand, using our method, the Frobenius norm becomes $||\hat{\boldsymbol{W}}_{hh}^{(\tau)} - \boldsymbol{W}_{hh}^{(\tau)}||_{\mathrm{F}}^2 = ||\boldsymbol{U\Sigma V} - \boldsymbol{U\bar{\Sigma}V}||_{\mathrm{F}}^2 = ||\boldsymbol{U}(\boldsymbol{\Sigma} - \boldsymbol{\bar{\Sigma}})\boldsymbol{V}||_{\mathrm{F}}^2 = \mathrm{tr}(\boldsymbol{V}^*(\boldsymbol{\Sigma} - \boldsymbol{\bar{\Sigma}})\boldsymbol{U}^*\boldsymbol{U}(\boldsymbol{\Sigma} - \boldsymbol{\bar{\Sigma}})\boldsymbol{V}) = \mathrm{tr}(\boldsymbol{V}^*(\boldsymbol{\Sigma} - \boldsymbol{\bar{\Sigma}})^2\boldsymbol{V}) = \mathrm{tr}(\boldsymbol{V}\boldsymbol{V}^*(\boldsymbol{\Sigma} - \boldsymbol{\bar{\Sigma}})^2) = \mathrm{tr}((\boldsymbol{\Sigma} - \boldsymbol{\bar{\Sigma}})^2) = \sum_{i=1}^{s}\{\sigma_i(\hat{\boldsymbol{W}}_{hh}^{(\tau)}) - (2-\delta)\}^2$, where $\mathrm{tr}(\cdot)$ is the trace of a matrix. As described above, the obtained $\boldsymbol{W}_{hh}^{(\tau)}$ satisfies $||\hat{\boldsymbol{W}}_{hh}^{(\tau)} - \boldsymbol{W}_{hh}^{(\tau)}||_{\mathrm{F}}^2 = \sum_{i=1}^{s}\{\sigma_i(\hat{\boldsymbol{W}}_{hh}^{(\tau)}) - (2-\delta)\}^2$, and this is the lower bound of eq. (13). Therefore, it is the solution of eq. (13). $\qquad\square$

### A.6 Proof of Lemma 6

**Lemma 6.** *The singular value of $\hat{\boldsymbol{W}}_{hh}^{(\tau)}$ is bounded as the following inequality:*

$$\sigma_i(\hat{\boldsymbol{W}}_{hh}^{(\tau)}) \leq \sigma_i(\boldsymbol{W}_{hh}^{(\tau-1)}) + |\eta|||\nabla_{\boldsymbol{W}_{hh}}C_{D_\tau}(\boldsymbol{\theta}))||_F. \qquad (15)$$

*Proof.* The following inequalities hold for any square matrices $\boldsymbol{M} \in \mathbb{R}^{q \times q}$ and $\boldsymbol{K} \in \mathbb{R}^{q \times q}$ [1]: $\sigma_{i+j-1}(\boldsymbol{M} + \boldsymbol{K}) \leq \sigma_i(\boldsymbol{M}) + \sigma_j(\boldsymbol{K})$, and $\sigma_1(\boldsymbol{K}) \leq ||\boldsymbol{K}||_F$. As a result, $\sigma_{i-1}(\boldsymbol{M} + \boldsymbol{K}) \leq \sigma_i(\boldsymbol{M}) + \sigma_1(\boldsymbol{K}) \leq \sigma_i(\boldsymbol{M}) + ||\boldsymbol{K}||_F$. Therefore, $\sigma_i(\hat{\boldsymbol{W}}_{hh}^{(\tau)}) = \sigma_i(\boldsymbol{W}_{hh}^{(\tau-1)} - \eta\nabla_{\boldsymbol{W}_{hh}}C_{D_\tau}(\boldsymbol{\theta}))) \leq \sigma_i(\boldsymbol{W}_{hh}^{(\tau-1)})) + \sigma_1(-\eta\nabla_{\boldsymbol{W}_{hh}}C_{D_\tau}(\boldsymbol{\theta}))) = \sigma_i(\boldsymbol{W}_{hh}^{(\tau-1)})) + |\eta|\sigma_1(\nabla_{\boldsymbol{W}_{hh}}C_{D_\tau}(\boldsymbol{\theta}))) \leq \sigma_i(\boldsymbol{W}_{hh}^{(\tau-1)}) + |\eta|||\nabla_{\boldsymbol{W}_{hh}}C_{D_\tau}(\boldsymbol{\theta}))||_F. \qquad\square$

## B  Preliminary Experiments

We conducted preliminary experiments in order to select the learning methods (such as Adam [4] or RMSprop [6]) and tune the hyper-parameters. In these experiments, we evaluated the model for each experimental condition. The model architectures were as the same as in the main experiments in the paper. The number of epochs was set to 60 in the language modeling experiment. In the music modeling experiments of Adam and RMSprop, we stopped the learning process if we observed no improvement of over 50 consecutive epochs. The experimental condition of the music modeling experiment of SGD is the same as in the main experiment. We evaluated the validation losses at each epoch, and we compared the methods using their lowest losses during their learning processes. Thresholds of gradient clipping were set to 5 in language modeling and 15 in music modeling because the average norms of the gradient in language modeling and music modeling were 10 and 30, respectively. In the language modeling experiment of SGD with the learning rate 0.1, we evaluated thresholds [5, 10, 15, 20] and delta of our methods [0.6, 0.9, 1.2, 1.5, 1.8]. This is because the small threshold degraded performance when the small learning rate was used. In addition, we evaluated success rates because we expected that the small learning rate might alleviate the gradient exploding problem.

### B.1  Learning Method and Range of Hyper-parameters

Learning methods and their hyper-parameters we examined were as follows:

- Adam: $\alpha \in \{10^{-2}, 10^{-3}, 10^{-4}\}$, $\beta_1 = 0.9$, $\beta_2 = 0.999$, $\epsilon = 10^{-8}$.
- RMSprop: Learning rate $\eta \in \{10^{-2}, 10^{-3}, 10^{-4}\}$, $\beta = 0.99$, $\epsilon = 10^{-8}$.
- SGD: We set learning rate $\eta$ to 1.0 and we divided $\eta$ by $\{1.2, 1.1, 1.05\}$ for each epoch after 10 epochs in language modeling on the basis of the previous study [9]. We also evaluated learning rate 0.1 without decay. In music modeling, we set $\eta$ to 0.1 and divided $\eta$ by 1.25 if we observed no improvement over 10 consecutive epochs following the previous study [3].

### B.2  Results

The lowest validation losses in the learning processes are listed in Tables 1 - 3. In Table 1, N/A means the validation perplexity diverged in the learning process. Tables 1 and 2 show the GRU trained by SGD ($\eta$ divided by 1.1) achieved the lowest perplexity. Therefore, we used SGD ($\eta$ divided by 1.1) in the main experiments in language modeling. In Table 3, Adam ($\alpha = 10^{-3}$) achieved the lowest loss among the results of our method. On the other hand, SGD achieved the lowest loss among the results of gradient clipping. SGD achieved the 2nd lowest loss among the results of our method. Since both

Table 1: Language modeling results

| Our method Method (hyper-parameters) | $\delta = 0.2$ Validation | Gradient clipping Method (hyper-parameters) | threshold = 5 Validation |
|---|---|---|---|
| Adam ($\alpha = 10^{-2}$) | 116.7 | Adam ($\alpha = 10^{-2}$) | N/A |
| Adam ($\alpha = 10^{-3}$) | 121.0 | Adam ($\alpha = 10^{-3}$) | 121.1 |
| Adam ($\alpha = 10^{-4}$) | 164.2 | Adam ($\alpha = 10^{-4}$) | 167.2 |
| RMSprop ($\eta = 10^{-2}$) | 129.0 | RMSprop ($\eta = 10^{-2}$) | N/A |
| RMSprop ($\eta = 10^{-3}$) | 116.0 | RMSprop ($\eta = 10^{-3}$) | 115.5 |
| RMSprop ($\eta = 10^{-4}$) | 166.5 | RMSprop ($\eta = 10^{-4}$) | 167.0 |
| SGD ($\eta$ divided by 1.2) | 105.5 | SGD ($\eta$ divided by 1.2) | 109.5 |
| **SGD ($\eta$ divided by 1.1)** | **103.5** | **SGD ($\eta$ divided by 1.1)** | **109.2** |
| SGD ($\eta$ divided by 1.05) | 105.2 | SGD ($\eta$ divided by 1.05) | 109.9 |

Table 2: Language modeling results (SGD $\eta = 0.1$)

| Our method Delta | 0.2 | 0.5 | 0.8 | 1.1 | 1.4 | Gradient clipping Threshold | 5 | 10 | 15 | 20 |
|---|---|---|---|---|---|---|---|---|---|---|
| Success Rate | 100 % | 100 % | 100 % | 100 % | 100 % | Success Rate | 50 % | 100 % | 100 % | 80 % |
| Validation | 138.1±0.8 | 137.9±0.7 | 137.4±0.6 | 137.0±0.6 | 136.8±0.4 | Validation | 178.8±0.8 | 154.7±0.8 | 145.1±0.8 | 219.3±1 |

Table 3: Music modeling results

| Our method Method (hyper-parameters) | $\delta = 0.2$ Validation | Gradient clipping Method (hyper-parameters) | threshold = 15 Validation |
|---|---|---|---|
| Adam ($\alpha = 10^{-2}$) | 4.37 | Adam ($\alpha = 10^{-2}$) | 13.7 |
| **Adam ($\alpha = 10^{-3}$)** | **3.38** | Adam ($\alpha = 10^{-3}$) | 3.73 |
| Adam ($\alpha = 10^{-4}$) | 10.8 | Adam ($\alpha = 10^{-4}$) | 12.9 |
| RMSprop ($\eta = 10^{-2}$) | 4.80 | RMSprop ($\eta = 10^{-2}$) | 13.9 |
| RMSprop ($\eta = 10^{-3}$) | 3.42 | RMSprop ($\eta = 10^{-3}$) | 3.56 |
| RMSprop ($\eta = 10^{-4}$) | 3.58 | RMSprop ($\eta = 10^{-4}$) | 3.63 |
| SGD | 3.42 | **SGD** | **3.55** |

Table 4: Language modeling results of gradient elementwise clipping

| Gradient clipping Threshold | 0.0015 | 0.003 | 0.0045 | 0.006 |
|---|---|---|---|---|
| Success Rate | 70 % | 10 % | 10 % | 10 % |
| Validation | 268.7.8±0.9 | 326.5 | 310.3 | 330.6 |
| Validation | 276.0±1.7 | 331.3 | 312.9 | 334.8 |

of our method and gradient clipping showed good performance, we used SGD (learning rate schedule was set as the previous study [3]) in music modeling.

## C  Gradient Elementwise Clipping

We evaluated gradient elementwise clipping in language modeling. No previous studies suggested guidlines for tuning a threshold of gradient elementwise clipping to the best of our knowledge. Therefore, we tuned the threshold based on the average value of the elements of the gradient in the same way as the gradient norm clipping. We varied the threshold in [0.0015, 0.003, 0.0045, 0.006] because the average value of the elements of the gradient was about 0.003. The other settings of the experiment is the same as that of gradient norm clipping.

The results of gradient elementwise clipping are listed in Table 4. We can see that perplexity and success rate of gradient elementwise clipping were lower than gradient norm clipping. This is because gradient elementwise clipping changes the direction of parameter updates as the results of clipping while gradient norm clipping does not change the direction. In addition, gradient elementwise clipping was more difficult to tune the threshold than that of gradient norm clipping because elementwise clipping uniformly clips elements of the gradient of which scales are different among the parameters.

## D    Conditions of Computation Time Experiment

The conditions of this experiment were the same as in the language modeling experiment using our proposed method ($\delta = 0.2$) and gradient clipping (threshold = 5). The experiment used the following set up: GPU: NVIDIA Tesla M40, CPU: Intel Xeon E5-2640 v4 2.40GHz, 1 TB of main memory, Ubuntu 16.04, CUDA (version 8.0) and cuDNN (version 5.1). The implementation of our model was based on Chainer (version 1.18.0) [7], and used NumPy (version1.12.1) for the naive SVD and fbpca[1] for the truncated SVD.

## Footnotes

[1]https://github.com/facebook/fbpca