[Reviews · NeurIPS 2017]

Reviewer 1



This paper addresses an important problem in training a class of recurrent neural networks: the Gated Recurrent Units (GRU). Training these important class of neural networks is not an easy problem and this paper makes an important contribution that throws light on the gradient explosion issue. A well written paper, the review in section 2 gives a succinct summary of the model and previous work. The proposed algorithm is well-motivated and clearly explained. I believe I can implement this immediately by following the steps. The discussion on computational cost is a little unclear, and the results shown in Fig. 2 is indeed disappointing (there really isn’t a saving, but the authors seem to give a spirited projection that this may be the case in section 4.4). Results shown in Tables 1 and 2 are important – they relate to success / otherwise of getting a network to learn. I think this is a problem easily hidden in a culture of only “reporting good news” – and that the gradient clipping method actually fails to find solutions is worthy of note.

Reviewer 2



Summary The authors propose a method for optimizing GRU networks which aims to prevent exploding gradients. They motivate the method by showing that a constraint on the spectral norm of the state-to-state matrix keeps the dynamics of the network stable near the fixed point 0. The method is evaluated on language modelling and a music prediction task and leads to stable training in comparison to weight clipping. Technical quality The motivation of the method is well developed and it is nice that the method is evaluated on two different real-world datasets. I also like the plots that show the relation between the spectral radius of the linear approximation of the system and the norm of the gradient. However, one important issue I have with the evaluation is that the learning rate is not controlled for in the experiments. Unfortunately, this makes it hard to draw strong conclusions from the results. The starting learning rates seem to be quite high and this may very well be the reason why the clipping-based optimization performs so poorly. Experiments with a range of different starting learning rates could give more insights into the importance of the method in general. Of course it would still be a strength of the method that it is less sensitive to the choice of learning rate but if simply lowering the learning rate a bit solves most of the issues the clipping has, it lessens the impact of the method. Clarity Despite the large number of equations, the argumentation is easy to follow. Except for some grammar mistakes, the paper is well-written in general. Novelty While recurrent neural networks have been quite thoroughly analysed in similar ways, this is to my knowledge the first work that does this for GRUs. I also don't know about any methods that combine SGD training with SVD to clip certain singular values. Significance While the analysis is specific to GRUs, I don't see why it couldn't inspire similar analysis of other popular RNN architectures. GRUs are also quite popular and for that reason the proposed method could have practical value for people. That said, it is hard to predict whether some new architecture will show up and take over or not and whether this method will be significant in the long run. pros: The method is well-motivated and the derivation can be useful by itself. Very insightful plot of the relation between the gradient norm and spectral radius. GRUs are very popular at the moment so if the method works well it may find widespread use. cons: Only works for GRUs The experiments could be the result of the particular choice of starting learning rate